# Determination of T-2 and HT-2 Toxins in Seed of Milk Thistle [*Silybum marianum* (L.) Gaertn.] Using Immunoaffinity Column by UPLC-MS/MS

**DOI:** 10.3390/toxins14040258

**Published:** 2022-04-06

**Authors:** Rastislav Boško, Marek Pernica, Sylvie Běláková, Marie Bjelková, Helena Pluháčková

**Affiliations:** 1Department of Crop Science, Breeding and Plant Medicine, Faculty of AgriSciences, Mendel University in Brno, Zemědělská 1, CZ-61300 Brno, Czech Republic; helena.pluhackova@mendelu.cz; 2Research Institute of Brewing and Malting, Mostecká 7, CZ-61400 Brno, Czech Republic; pernica@beerresearch.cz (M.P.); belakova@beerresearch.cz (S.B.); 3Agritec Plant Research, Zemědělská 2520/16, CZ-78701 Sumperk, Czech Republic; bjelkova@agritec.cz

**Keywords:** milk thistle, T-2 toxin, HT-2 toxin, validation method, immunoaffinity column, UPLC-MS/MS

## Abstract

Milk thistle [*Silybum marianum* (L.) Gaertn.] achieved a significant increase in interest over the past few years from local and foreign pharmaceutical corporations. The silymarin complex of constituents extracted from milk thistle achenes provides compelling health benefits primarily thanks to antioxidant activities and hepatoprotective effects. However, consuming mycotoxin-contaminated plant material can cause immunosuppression and hepatotoxic problems. The aim of this study was to develop and validate a method for the determination of mycotoxin content in milk thistle. Fusarium toxins as T-2 and HT-2 toxins in grown milk thistle harvested from a breeding station in the Czech Republic during 2020–2021 were studied. The analysis of T-2 and HT-2 toxins was performed by UPLC-MS/MS after immunoaffinity columns EASI-EXTRACT^®^ T-2 & HT-2 clean up. All analysed samples of milk thistle were contaminated with T-2 toxin and HT-2 toxin. The content of T-2 toxin in the samples from 2020 was in the range of 122.7–290.2 µg/kg and HT-2 toxin 157.0–319.0 µg/kg. In 2021, the content of T-2 toxin was in the range of 28.8–69.9 µg/kg and HT-2 toxin was 24.2–75.4 µg/kg. The results show that the climatic conditions of the year of harvesting have a highly statistically significant effect on the content of T-2 and HT-2 toxins in milk thistle.

## 1. Introduction

Milk thistle [*Silybum marianum* (L.) Gaertn.] is an annual, and sometimes biennial, medicinal plant of the family Asteraceae cultivated in agriculture, and it is commonly used as a medicinal drug [1,2]. In recent years, there has been a significant increase in demand from domestic and foreign processors, mainly from the pharmaceutical, cosmetics and feed industries. The milk thistle has a significant share in increasing the growing areas of medicinal plants and is the most cropped medicinal plant in the Czech Republic [1].

The subject of the crop are achenes—*Silybi mariani fructus* (Ph. Eur.)—and it has been used in health remedies since the time of ancient Greece [1,3]. The main beneficial constituent of milk thistle is silymarin—a complex of flavonolignans containing silybin A, silybin B, isosilybin A, isosilybin B, silydianin and silychristin—which exerts a positive effect on liver health [4,5,6]. Milk thistle also contains flavonoids—taxifolin and quercetin. Other potentially beneficial attributes of milk thistle include its antioxidant, antihypercholesterolemic and chemoprotective effects against lung and prostate cancer [7,8,9,10]. Milk thistle has anticancer activity, anti-inflammatory, immunomodulatory activity and neuroprotective potential [11].

The content of the silymarin complex depends on the variety of a milk thistle, e.g., the variety *Silyb* is approximately 2% and the minimum content according to the pharmacopoeia is 1% of silymarin complex per kilo of drug [1].

With the recent natural therapy revolution, the consumption of milk thistle supplements has widely spread among consumers in Europe [12]. Herbal medicine has been increasingly used for therapeutic purposes against a diverse range of human diseases worldwide. However, contaminants in herbal medicines produce severe problems that have seriously affected the value of herbal products and damaged human health [13].

As one of the major contaminants, mycotoxins are the secondary metabolites produced by various species of filamentous fungi, such as *Aspergillus*, *Penicillium*, *Fusarium* or *Alternaria*, and trigger several ailments of the kidneys, liver, skin and respiratory system. Among mycotoxins, aflatoxins, fumonisins, ochratoxin A, zearalenone and trichothecenes such as deoxynivalenol, T-2 toxin or HT-2 toxin are the most frequently detected mycotoxins in herbal medicines [14,15,16,17]. Generally, mycotoxins can develop either in the pre-harvest, post-harvest or storage stages. Climate change, poor storage and damage from harvest processing make crops more susceptible to mycotoxin contamination. In particular, milk thistle is characterized by the non-uniform ripening time of an individual flower head on the plant, which represents ideal conditions for the growth of fungi [15,16,17].

Mycotoxins have a different chemical structure, on which their metabolism and degree of toxicity in the organism depend. The genus *Fusarium* is one of the toxicogenic fungi with a high production of various mycotoxins. The optimal conditions for the growth and production of mycotoxins by microscopic fibrous fungi of the genus *Fusarium* are: temperature 15–25 °C, relative humidity 20–25%, water activity 0.98–0.99 and weakly acidic pH of the environment. The three most important groups of mycotoxins synthesized by the genus *Fusarium* are: trichothecenes, zearalenone and fumonisins [18,19].

The *Fusarium* toxins such as T-2 toxin, HT-2 toxin, diacetoxyscirpenol or neosolaniol are classified as type A trichothecenes [20]. Their main producers are *Fusarium sporotrichioides*, *F. poae*, *F. langsethiae* or *F. acuinatum.* As dominant field fungi in Europe are widely held in cereal as natural contaminants or other plants, e.g., medicinal plants including milk thistle [9,21,22,23,24].

The content of mycotoxins in milk thistle has been analysed in several studies. *Fusarium* mycotoxins such as T-2 and HT-2 toxins, zearalenone, deoxynivalenol, 3-acetyldeoxynivalenol, fusarenon-X, diacetoxyscirpenol, neosolaniol or enniatins and beauvericins have been found in the samples, as well as high levels of *Alternaria* mycotoxins such as alternariol, alternariol-methyl ether, tentoxin or tenuazonic acid [14]. Fumonisins, aflatoxins, citrinin have been detected in several medicinal plants, including milk thistle [25]. Nivalenol, ochratoxin A, i.a., have been determined by Arroyo-Manzanares et al. in a multiclass mycotoxin analysis [26]. Fenclova [27] further analysed sterigmatocystin or mycophenolic acid.

Many analytical methods have been published concerning the determination of mycotoxins in different matrices. In general, chromatographic methods such as liquid chromatography coupled with mass spectrometry is the most commonly used technique for mycotoxin analysis, with other analytical techniques being seldom used due to their limited sensitivity and selectivity [28].

T-2 toxin, which belongs to the type A trichothecenes, has one of the highest toxicities of all the trichothecenes [29]. T-2 toxin is often produced by different *Fusarium* species which may grow on a variety of grains or seeds, especially in cold climate regions or during wet storage conditions [29,30]. In addition to fungal genetics, T-2 toxin production can be significantly influenced by environmental conditions, such as temperature, humidity and the type of substrate [31]. T-2 toxin is non-volatile and resistant to degradation in different environments, such as light and temperature, but it is effectively deactivated by strong acid or alkaline conditions. T-2 toxin causes a large range of toxic effects in animals, such as weight loss, decreases of blood cell and leukocyte count, reduction in plasma glucose and pathological changes in the liver and stomach. Furthermore, T-2 toxin is associated with an increased infection rate and induction of apoptosis [32,33,34,35,36,37,38]. T-2 toxin was considered as the cause of alimentary toxic aleukia (ATA) or Kashin–Beck disease [39]. Toxic effects of these mycotoxins are well documented in animals. In particular, T-2 toxin is a potent inhibitor of protein, deoxyribonucleic acid (DNA) and ribonucleic acid (RNA) synthesis, both in vitro and in vivo, and has immunosuppressive and hematotoxic effects [39,40,41].

T-2 toxin is mainly metabolized into HT-2 toxin, a deacetylated form of T-2 toxin, which has comparable toxicity with T-2 toxin (Figure 1). A small amount of T-2 toxin can also be metabolized into diacetoxyscirpenol and/or neosolaniol [42,43,44,45,46]. The EFSA Panel on Contaminants in the Food Chain (CONTAM) established a tolerable daily intake (TDI) for T-2 toxin and HT-2 toxin of 0.02 μg/kg body weight per day for the sum of T-2 and HT-2 toxins [47].

The aim of this study was to develop and validate the analytical UPLC-MS/MS method using EASI-EXTRACT^®^ T-2 & HT-2 immunoaffinity columns for the determination of T-2 and HT-2 toxins in milk thistle samples. The effect of row width (individual variants) was investigated (*n* = 4) to determine the effect of edge position in the plot of the randomized experiment. The effect of the climatic conditions of the year on the content of selected mycotoxins was also investigated.

## 2. Results

### 2.1. Method Validation

The analytical method was validated for selected parameters such as linearity, limit of detection (LOD), limit of quantification (LOQ), precision and accuracy. Calibration curves of the selected mycotoxins were linear in the working range of LOQ–1000 ng/mL. The coefficients of determination (R^2^) were 0.9998 and 0.9999 for T-2 and HT-2 toxin, respectively; the regression equations were (y = 3298x − 1339.1) and (y = 1047.4x − 669.79) for T-2 and HT-2 toxins, respectively.

Detection and quantification limits were determined using consecutive dilution until the signal-to-noise ratio became unsuitable, and the concentrations were calculated as the ratio of S/N = 3 and S/N = 10 for LOD and LOQ, respectively. For T-2 toxin, LOD 0.04 μg/kg and LOQ 0.1 μg/kg were determined, for HT-2 toxin, LOD was 0.25 μg/kg and LOQ 0.8 μg/kg. The precision and accuracy for T-2 and HT-2 toxins were verified at three concentration levels; 25 μg/kg, 100 μg/kg and 500 μg/kg; triplicate. Recoveries for T-2 toxin varied in the range 96.7–106.9% with RSDs ≤ 7.6% (*n* = 3) and for HT-2 in the range 71.6–78.8% with RSDs ≤ 9.9% RSD (*n* = 3).

Validity of the method was also verified via interlaboratory comparative trials organized by Central Institute for Supervising and Testing in Agriculture (ÚKZÚZ Brno, the Czech Republic). Six laboratories successfully participated in the testing for T-2 toxin and HT-2 toxin in ground milk thistle seeds with standard z-score −1.53–1.40 and −1.46–1.02, respectively. Obtained values varied in the range 39.3–77.4 μg/kg and 34.5–62.2 μg/kg, respectively; the average values were 59.1 μg/kg and 50.8 μg/kg, respectively.

### 2.2. Determination of T-2 and HT-2 Toxins in Milk Thistle by UPLC-MS/MS

The content of T-2 toxin in the samples from 2020 was in the range of 122.7–290.2 µg/kg and HT-2 toxin 157.0–319.0 µg/kg. The average sum of T-2 and HT-2 toxins was 389.4 µg/kg. In 2021, the content of T-2 toxin was in the range of 28.8–69.9 µg/kg and HT-2 toxin was 24.2–75.4 µg/kg. The average sum of T-2 and HT-2 was 97.5 µg/kg. The obtained results indicate that the content of T-2 and HT-2 toxins was statistically significantly higher in samples from the harvest year 2020, compared to the harvest year 2021. The results of this study are shown in Table 1 and Table 2. The T-2 toxin content was statistically significantly highest in the 2020 samples in Variant 12.5 B: 284.8 µg/kg. In contrast, the statistically significantly lowest content was found in the samples of Variant 25 C: 124.3 µg/kg. However, these samples were not statistically significantly different from the samples of Variant 25 A: 125.4 µg/kg. In 2020, the statistically significantly highest HT-2 toxin content was found in the samples of Variant 37 D: 313.9 µg/kg. The statistically significantly lowest HT-2 toxin content in 2020 was found in the samples of Variant 25 C: 157.8 µg/kg.

In the year 2021, the statistically significantly highest T-2 toxin content was found in the samples of Variant 12.5 B: 69.5 µg/kg. These results were not statistically significantly different from the samples of Variant 37 C: 65.8 µg/kg. In contrast, the statistically significantly lowest T-2 toxin content was found in the samples of Variant 25 A and D: 29.2 and 33.3 µg/kg, respectively, which were not statistically significantly different from the samples of Variant 12.5 A: 33.8 µg/kg. The statistically significantly highest HT-2 toxin content was detected in the samples of Variant 25 D: 87.7 µg/kg. On the other hand, the statistically significantly lowest HT-2 content was also detected in the samples of Variant 25 A: 25.0 µg/kg. However, these results were not statistically significantly different from the samples of Variant 37 A: 28.1 µg/kg. Table 2 shows that the majority of the 2020 samples were found to have higher levels of HT-2 toxin, with the exception of the Variant 12.5 B sample, which had higher levels of T-2 toxin. Compared to 2020, in the year 2021, the T-2 toxin content was higher in more samples, especially in Variant 25 A, B and C (29.2, 58.2 and 56.9 µg/kg, respectively). Sample D of the same Variant 2 showed several times higher HT-2 toxin content (87.7 µg/kg) compared to T-2 toxin content (33.3 µg/kg). At the same time, the HT-2 toxin content was statistically significantly the highest among all the samples from the year 2020. Furthermore, the T-2 toxin content was higher in the samples of milk thistle Variant 37 A, B, C and D (44.9, 39.0, 65.8 and 47.9 µg/kg, respectively).

The analysis of variance showed that the effect of year on T-2 and HT-2 mycotoxin content in milk thistle samples in this study was statistically very highly significant, as well as the effect of individual variants of the randomized trial with different row widths (*n* = 4 for each Variant). From subsequent testing, it can be concluded that the repetition that was located in the position at the plot edge could have shown higher mycotoxin content.

## 3. Discussion

Milk thistle has been examined for mycotoxin contamination and reports of research carried out showed some cases of significant levels of contamination, in particular by aflatoxins, deoxynivalenol, T-2 and HT-2 toxins, exceeding the allowed maximum limits. The results were primarily found in dietary supplements [9,12,14,25,26,27,48].

Santos et al. [25] presented a study of medicinal and aromatic plants obtained from Spain, monitored for possible mycotoxin contamination, including two milk thistle samples, both contaminated by T-2 toxin (17.5 µg/kg and 35.6 µg/kg). HT-2 toxin has not been analysed. Arroyo-Manzanares et al. [26] developed a multiclass QuEChERS method for the analysis of mycotoxins in milk thistle. Different samples of commercial milk thistle purchased in local markets from Spanish Granada were analysed to show the applicability of the method; two samples gave positive results for T-2 toxin (363.0 µg/kg and 453.9 µg/kg) and HT-2 toxin (826.9 µg/kg and 943.7 µg/kg). Veprikova et al. [14] developed a multiclass QuEChERS method for 57 various mycotoxins for herb-based supplements. In total, 32 milk thistle-based supplements from Czech and US retail markets for treatment of liver diseases were examined. The level of maximal concentration of T-2 toxin was 1870 µg/kg and HT-2 toxin 1530 µg/kg. Highest total mycotoxin content found in milk thistle-based supplements reached 37 mg/kg. Fenclova et al. [27] studied the composition, chemical and biological safety of 26 milk thistle-based dietary supplements purchased from United States and Czech Republic markets in 2016–2017. The most frequent mycotoxins found in the milk thistle-based preparations were the *Fusarium* mycotoxins. Type A trichothecenes, T-2 and HT-2, toxins occurred in 92% and 96% of the samples, respectively. Pickova et al. [9] monitored the content of mycotoxins in milk thistle-based supplements; T-2 toxin was detected in 52 of 67 food supplement samples. The highest amounts of T-2 toxin were found in capsules with dried powder (5958 µg/kg) and seeds (453.9 µg/kg). HT-2 toxin was present in 48 of 65 dietary supplement samples. The maximum levels of HT-2 toxin were found in capsules with dried powder (2985 µg/kg) and seeds (943.7 µg/kg).

As previously reported, milk thistle achenes tend to contain not only bioactive substances favourable for human and animal health, such as the silymarin complex, but also naturally occurring contaminants: mycotoxins produced as secondary metabolites by microscopic filamentous micromycetes [27]. The occurrence of mycotoxins in food and feed is still a problem of utmost importance [49]. Higher levels of mycotoxins in raw materials for the pharmaceutical industry or the production of dietary supplements may be influenced by several factors. In the case of milk thistle, indeterminate growth is problematic in terms of cultivation technology, as it causes uneven flowering and maturation of seeds on the plant [50]. It is rather common for milk thistle that mature seeds fall out from the anthodia prematurely or are eaten by birds. Proper timing of the harvest is therefore crucial, and it is quite difficult to determine because of the uneven ripening of the achenes [50,51]. Soil and climate conditions during cultivation are another important factor in the development of fusariosis or other spread of other mycotoxin producers [51,52]. Inappropriate post-harvest treatment can be another undesirable factor, especially insufficient drying; after harvesting the achenes must be cleaned and dried at 50 °C to moisture content ≤ 8 % [51]. The presence of mycotoxins could be controlled by cultivation technologies; however, the use of pesticides is not allowed for this commodity as it is not in a good accordance with a current trend of global pesticide use reduction. In this study, only herbicide treatments were used early in the development of milk thistle plants. Another very important task is increase the awareness of milk thistle growers to mycotoxin producers to encourage them to pay more attention to the quality of raw materials. The occurrence of mycotoxin pathogens can be significantly influenced by the configuration of cultivation technologies [53].

In the European Union, maximum limits have not been set yet for the mycotoxin content in milk thistle for pharmaceutical industry or food supplements. However, the concentrations determined in this work indicate that the values often exceed maximum limits set for other commodities, such as cereals. According to the standard EC 165/2013 the legally recommended maximum levels for the sum of T-2 and HT-2 toxins in cereals intended for human consumption is 200 μg/kg [54]. The sum of T-2 and HT-2 toxins in Sample 12.5 B from 2020 (avg. 562.4 μg/kg) exceeded the limit 2.8-fold. This comparison of mycotoxin concentrations determined in milk thistle samples with maximum limits given for cereals is of course purely theoretical, as the maximum limits take into account not only toxicity but also the amount of food that is normally consumed [14]. The best way to express the risk associated with mycotoxins in milk thistle is to calculate the daily exposure based on the recommended dosage and compare such calculated value with the tolerable daily intake (TDI) given by the regulations. The daily intake of mycotoxins calculated on the basis of the recommended food consumption should be considered to assess the potential health risk associated with the occurrence of mycotoxins. Highest daily exposure is due to the consumption of contaminated milk thistle achenes [14,27]. The most common use of milk thistle is direct oral consumption in the form of ground achenes, standardized tablets, capsules or extracts containing 200–400 mg of silymarin complex. Consumption of milk thistle in the form of tea is highly discouraged because silymarin is only slightly soluble in water [9,55]. Recommended daily dose of freshly ground seeds is up to 15 g (2 tablespoons) per day [55] and the TDI has been set at 0.02 μg/kg body weight per day for the sum of T-2 and HT-2 toxins [47]. Thus, the uptake of the T-2 and HT-2 toxins can reach up to 600% TDI for a person that weights 70 kg (in Sample 12.5 B from 2020, sum of T-2 and HT-2 toxins was 562.4 μg/kg). For the least contaminated Sample 25 B from 2021, containing only 54.2 µg/kg, the uptake reached 80% of TDI. Consumption of Samples 12.5 C and 37 D from the 2021 harvest would cause 130% of TDI. Consumption of such contaminated seeds is highly undesirable as T-2 toxin and its non-acetylated form HT-2 toxin show immunotoxic or hepatotoxic effects on the organism. The toxic impact may be even more severe if dietary supplements of multiple plant constituents with high concentrations of several mycotoxins are consumed, where a so-called mycotoxin “cocktail” is formed [14].

Based on the recommended dosage, none of the monitored samples from the 2020 harvest would have met the limit; however, this could be caused by atypical rainfall [56]. Published data shows that certain weather conditions favour the growth of *Fusarium* moulds and that contamination of milk thistle with *Fusarium* mycotoxins is possible, especially during rainy and warmer periods [53,57,58,59]. Figure 2 shows that in 2020, the rainfall was intense mainly in February, June and August (96.2 mm, 136.5 mm and 164.5 mm). In August, before harvest, average rainfall was relatively high, which was unfavourable for harvesting techniques. Figure 3 shows that in in 2021, average rainfall was more similar to the long-term average compared to 2020, except for September and October when rainfall decreased significantly between August (90.3 mm) and September (17.5 mm) in and October (10.7 mm) it was 35 mm less than the long-term average. The average temperature in 2020 was higher than the long-term mean value, with a slight exception in May. The results show that from winter to spring (January–April) the temperatures were higher than the long-term average. In August and October, temperatures were about 2 °C higher. In 2021, the measured temperatures followed the long-term average curve, except for June and July, when temperatures were also higher than in 2020 (avg. 18.7 °C and 19.6 °C).

## 4. Materials and Methods

### 4.1. Standards and Chemicals

Solvent purity LC-MS grade as methanol (MeOH), acetonitrile (ACN), 2-propanol (IPA), acetone (ACE), dimethylsulfoxide (DMSO) and standards of mycotoxins T-2 toxin and HT-2 toxin were purchased from Sigma-Aldrich (Prague, Czech Republic). The stock solution standards of mycotoxins (100 µg/mL in acetonitrile) were prepared in 10% methanol/water (*v*/*v*) at a concentration of 1 µg/mL and stored at 4 °C in the refrigerator before use. Sodium chloride was purchased from Lachner (Neratovice, Czech Republic). Additives for LC-MS (purity LC-MS) as ammonium acetate, formic acid, acetic acid and citric acid (purity ≥ 99.5%) were purchased from Sigma-Aldrich (Prague, Czech Republic). Check-Sample-Survey, milk thistle powder obtained from Central Institute for Supervising and Testing in Agriculture (ÚKZÚZ Brno, Czech Republic). Ultrapure water was produced by Aqua Osmotic 06 (Tišnov, Czech Republic). Immunoaffinity columns R-Biopharm EASI-EXTRACT^®^ T-2 & HT-2 were obtained from Jemo Trading (Bratislava, Slovak Republic).

### 4.2. Samples Collection

Randomized milk thistle cultivation experiment was established in experimental station Agritec Plant Research Plc. in Šumperk in the Czech Republic in 2020 and 2021. Individual cultivation variants were carried out with different row widths—12.5 cm, 25 cm and 37 cm with *n* = 4 (A, B, C and D). The amount of sown seeds was 8 kg/ha. Sowing dates were 25 April 2020 and 30 April 2021. The crops were treated two times with registered herbicides, active ingredients were ethofumesate and in one case also quizalofop-*p*-ethyl. First treatments were carried out on 22 May 2020 and 25 May 2021, respectively, in the phase of 2 true leaves, second treatments were done on 1 June 2020 and 5 June 2021, respectively, in rosette phase. Harvest times were 9 September 2020 and 25 August 2021. The yields of achenes were 0.703 t/ha and 1.194 t/ha in 2020 and 2021, respectively. Average rainfall in 2020 was 76.53 mm and in 2021 it was 52.11 mm. Average temperatures were 10.42 °C and 9.24 °C, respectively. Meteorological data was obtained from meteorological station located in Agritec Plant Research Plc., Šumperk, Czech Republic (49°58′26.8″ N 16°58′03.2″ E).

A total of 24 milk thistle [*Silybum marianum* (L.) Gaertn.] seed samples of the variety *Mirel* with a weight of 150 g were taken from each sample. *Mirel* is a legally protected variety, the rights holder is Moravol Ltd. This variety has a high content of good quality oil with a specific fatty acid composition. It is used mostly for the isolation of silymarin complex, but the oil is also pressed from the seeds as a by-product.

The experiment was harvested with a small parcel harvester Sampo Rosenlew SR2010. Due to unfavourable weather conditions a large amount of moisture-increasing impurities was present in the harvested product at the time of harvest. Because of that, pre-cleaning was carried out immediately after the harvest and the harvested material was subsequently transferred to cold air drying.

The samples were cleaned and stored in paper bags in the dark under laboratory conditions: 22 °C and 42% relative humidity.

### 4.3. Sample Preparation and Clean-Up by Immunoaffinity Columns

Twenty-five grams of milk thistle, finely ground by a grinder AR 1105 (Moulinex, France), were weighed into a Duran Youtility^®^ bottle (Sigma-Aldrich, Prague, Czech Republic) with five grams of sodium chloride, added 125 mL 90% methanol/water (*v*/*v*) and shook for 50 min on orbital shaker IKA KS 260 (VWR, Stříbrná Skalice, Czech Republic). After centrifugation at 4800 rpm for 10 min, 20 mL of extract was diluted with 80 mL 2% sodium chloride and left on the bench for 5 min with occasional gently shaking the solution to allow precipitation to occur. Supernatant was filtered through Whatman No. 4 filter paper. The 25 mL aliquot of diluted filtrate was passed through the immunoaffinity column EASI-EXTRACT^®^ T-2 & HT-2, which was inserted into the Supelco SPE Vacuum Manifold Visiprep™ (Sigma-Aldrich, Prague, Czech Republic) at a gravitation flow rate. Slowly and steady flow rate is essential for the capture of the toxins by the antibody. Column was washed by passing 20 mL of water through at a flow rate approximately 5 mL/min, then passed air though the column to remove residual liquid. The toxins were eluted from the column at a flow rate of 1 drop per second using 2 mL 100% methanol and collected in a heart shaped flask. This eluate was evaporated to dry under vacuum in rotary evaporator IKA RV 10 (VWR, Stříbrná Skalice, Czech Republic). The residues were dissolved in 1 mL of 10% methanol/water (*v*/*v*) for UPLC-MS/MS analysis according to validated method (*n* = 2).

### 4.4. UPLC-MS/MS Mycotoxin Analysis

The Waters Acquity™ H-Class Plus UPLC^®^ system coupled to the Xevo TQ-S Micro mass spectrometer (Waters, Prague, Czech Republic) with electro-spray ionization (ESI) was used for the identification and quantification of mycotoxins. Data acquisition and processing were performed with MassLynx™ version 4.2 software (Waters, Prague, Czech Republic). Chromatographic separation was performed on a column Waters Acquity™ UPLC^®^ BEH C18 (100 × 2.1 mm, 1.7 µm) and the column temperature was maintained at 40 °C using gradient elution. The mobile phase consisted of eluent A (1 mM ammonium acetate + 0.5% acetic acid + 0.1% formic acid in water) and eluent B (0.5% acetic acid + 0.1% formic acid in methanol). The gradient program was applied at a flow rate of 0.4 mL/min under the following conditions: 0 min 90% A; 3.0 min 90% A; 10.0 min 30% A; 10.1 min 10% A; 12.0 min 10% A; 13.1 min 90% A; 15.0 min 90% A. The injection volume was 5 µL and injection mode was a flow-through needle. The needle wash consisted of 7.44 mM citric acid in water:MeOH:ACN:IPA:ACE:DMSO (37:9:19:19:9:7), respectively. Seal wash and purge wash was 10% IPA. The mass spectrometer was operated at ESI and parameters were set up as follows: capillary voltage 30 kV, source temperature and desolvation temperature were 120 °C and 450 °C, respectively. Collision gas was argon and nitrogen applied as spray gas: cone gas flow 100 L/Hr, desolvation gas flow 800 L/Hr. For selectivity, the mass spectrometer was operated in MRM mode and two transitions per mycotoxin were monitored.

### 4.5. Development, Optimization and Validation of Method

This section describes development, optimalization and validation the data generated by an Acquity™ H-Class plus with a Xevo TQ-S Micro mass spectrometer system for the analysis of T-2 toxin and HT-2 toxin using electrospray ionization. A UPLC method was created to analyse monitored compounds in a 15 min gradient and two MRM transitions (multiple reaction monitoring) were optimized for each analyte using positive or negative electrospray ionization (ESI). The optimized MRM parameters are shown in Table 3.

Linearity, limit of detection (LOD), limit of quantification (LOQ), precision and accuracy of the method were evaluated. Linearity was evaluated using standard solution in seven concentration levels. The LODs were determined as the lowest detectable mycotoxin concentration at which a signal-to-noise ratio (S/N) = 3. The LOQs were determined as the minimum amount of analyte for quantification with a S/N = 10. The accuracy and precision for mycotoxin compounds were determined at three concentration levels for three times. The calibration curves were linear with a good linearity where the coefficient of determination (R^2^ > 0.999) for each analyte.

### 4.6. Statistical Analysis

The obtained results were evaluated by one-way analysis of variance (ANOVA) followed by testing at a 95% (*p* < 0.05) level of significance using the Fisher LSD test. The data were processed using the STATISTICA.CZ version 12.0 software (StatSoft, Prague, Czech Republic). Results are expressed as a mean value ± standard deviation (SD).

## 5. Conclusions

The safety and quality of food and food raw materials is currently a highly discussed topic. Especially in crops that should bring health benefits, the content of undesirable compounds, harmful to human health, should be controlled. Milk thistle is an important medicinal plant, used mostly in the pharmaceutical industry, and its bioactive compounds help the detoxication of an organism. Thus, especially in these conditions, the occurrence of toxins in such a commodity is extremely alarming. The content of unwanted toxic compounds can be regulated by good agricultural practice and post-harvest treatment. However, the current situation is further complicated by the fact that there is no pesticide preparation effective against these mycotoxin producers legally registered at the moment. The toxicological impact of many mycotoxins that can be found in plants is still unknown, especially when they are present in mixtures, known as “mycotoxin cocktails”; thus, it is important to continue their further monitoring.

Due to the amount of T-2 and HT-2 toxins in some milk thistle samples exceeding 600% of the TDI, it is clear that high health risks can be expected, especially for people with weakened immune systems or liver diseases. Although it is not possible to completely eliminate mycotoxins from milk thistle or other plants, it would be advisable to set their maximum limits. For oilseeds, which include milk thistle, no limits have yet been incorporated into legislation, but standard limit values are already legislatively given for cereals, for example.

## Figures and Tables

**Figure 1 toxins-14-00258-f001:**
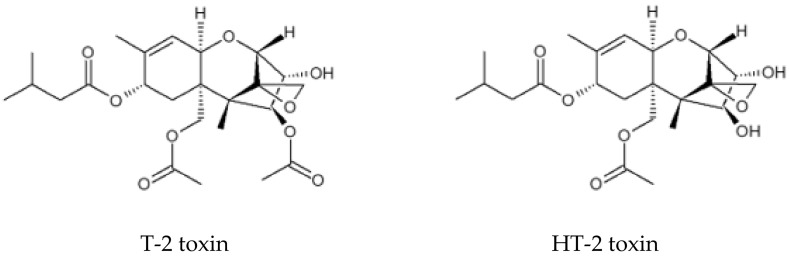
Chemical structure of trichothecenes T-2 and HT-2 toxins.

**Figure 2 toxins-14-00258-f002:**
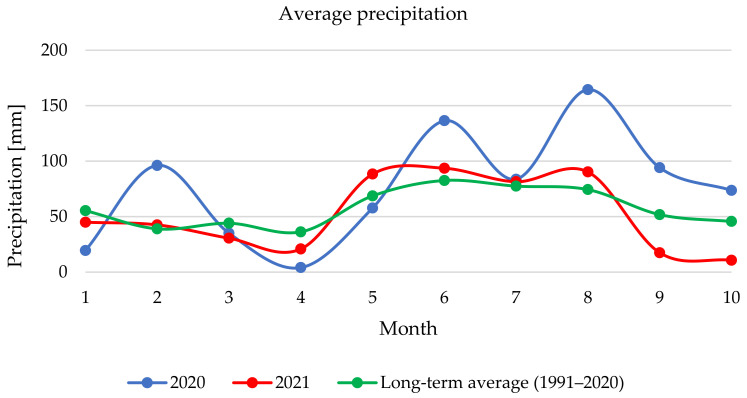
Average precipitation in Šumperk (Czech Republic) during vegetation and harvest period.

**Figure 3 toxins-14-00258-f003:**
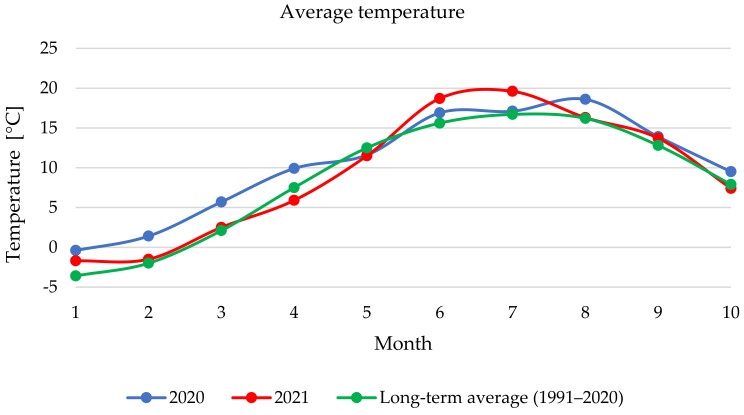
Average temperature in Šumperk (Czech Republic) during vegetation and harvest period.

**Table 1 toxins-14-00258-t001:** ANOVA for the average T-2 and HT-2 toxin content in milk thistle samples from 2020 and 2021.

Source of Variability	DF	T-2 Toxin [µg/kg] MS	HT-2 Toxin [µg/kg] MS
Year	1	194,669.8 ***	317,040.4 ***
Variant	11	2938.6 ***	2303.2 ***
Year*variant	11	1972.0 ***	2540.0 ***
Error	24	5.8	18.3

Legend: DF = Degree of Freedom, MS = Mean Square; * *p* ≤ 0.05; *** *p* ≤ 0.001.

**Table 2 toxins-14-00258-t002:** The average T-2 and HT-2 toxins content in samples from 2020 and 2021.

Year	Variant		T-2 Toxin [µg/kg]		HT-2 Toxin [µg/kg]	
2020	1. (12.5 cm)	A	180.3 ± 2.9	j	238.9 ± 3.4	o
		B	284.8 ± 7.7	n	277.6 ± 8.4	p
		C	136.5 ± 4.1	h	172.9 ± 1.2	k
		D	200.2 ± 1.5	l	204.8 ± 6.5	n
	2. (25 cm)	A	125.4 ± 1.3	g	187.8 ± 5.8	lm
		B	136.5 ± 0.2	h	188.9 ± 4.6	lm
		C	124.3 ± 2.2	g	157.8 ± 1.2	j
		D	160.2 ± 5.3	i	186.6 ± 6.0	l
	3. (37 cm)	A	176.1 ± 0.3	j	239.2 ± 9.8	o
		B	187.1 ± 0.7	k	201.4 ± 5.1	n
		C	158.9 ± 3.2	i	196.6 ± 1.7	mn
		D	236.1 ± 1.8	m	313.9 ± 7.3	q
2021	1. (12.5 cm)	A	33.8 ± 0.0	a	38.4 ± 1.1	bc
		B	69.5 ± 0.6	f	73.2 ± 3.1	h
		C	42.0 ± 0.2	bc	48.7 ± 1.3	de
		D	57.4 ± 0.5	e	62.9 ± 1.1	fg
	2. (25 cm)	A	29.2 ± 0.4	a	25.0 ± 1.2	a
		B	58.2 ± 0.4	e	54.7 ± 2.6	ef
		C	56.9 ± 0.2	e	53.7 ± 1.6	de
		D	33.3 ± 0.3	a	87.7 ± 2.5	i
	3. (37 cm)	A	44.9 ± 1.5	cd	28.1 ± 1.8	a
		B	39.0 ± 0.3	b	33.5 ± 1.5	ab
		C	65.8 ± 0.7	f	64.7 ± 2.6	gh
		D	47.9 ± 0.2	d	45.3 ± 0.4	cd

Legend: The average values marked with different letters in the rows differ statistically significantly at *p* = 0.05.

**Table 3 toxins-14-00258-t003:** Optimized MRM parameters for monitored mycotoxins.

Mycotoxin	Retention Time [min]	Precursor Ion [m/z]	Product Ion [m/z]	Collision Energy [V]
T-2 toxin	10.66	[M + NH_4_]^+^	185.00 *	22
		484.25	215.15 **	22
HT-2 toxin	10.03	[M + NH_4_]^+^	215.13 *	12
		442.17	263.15 **	12

* quantification ion; ** qualifier ion.

## Data Availability

Data are available upon request, please contact the contributing authors.

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
