# Peer review of "Determination of T-2 and HT-2 Toxins in Seed of Milk Thistle [Silybum marianum (L.) Gaertn.] Using Immunoaffinity Column by UPLC-MS/MS"

_toxins, 2022, doi:10.3390/toxins14040258_

Round 1
Reviewer 1 Report
The Authors determined occurrence of T-2 and HT-2 mycotoxins in seeds of milk thistle. The manuscript is of importance in viewpoint of pharmaceutical industry and possible contamination of medicals which are based on extract from milk thistle achenes. However, some part of the paper have to be corrected. Details are listed below:
First of all, I would like to ask why only T-2 and HT-2 toxins were analyzed?
L17-18: probably it is not the impact of the year itself but climatic conditions of the years (temperature and precipitation). Please rephrase
L22-23: see L17-18 comment
L68-72: try to give broader description of different mycotoxins determined in milk thistle by other researchers
L94: amount instead proportion
L100: indicate clearly on Figure 1, which structure is T-2 and which HT-2
L101-103: this part should be included to Materials and methods
L107-108: see L17-18 comment
L131-135: this part should be included to Materials and methods
L142: were shown
L157: Table 3 is related to MRM parameters, not mycotoxins concentration
L162-163: higher concentration of HT-2 was in 2020, not 2021.
L166: T-2 and HT-2 instead T2 and HT2. Also it would be more accurate to include temperature and precipitation from 2020 and 2021 for ANOVA in Table 1
L169-172: also it would be more accurate to make correlations between T-2, HT-2 and precipitation, and temperature
L174: remove the sentence
L178: this part should be added to the Discussion. Also, it should be more fluently written between further paragraphs
L183-184: remove or keep for conclusions
L203: Fusarium in italics
Figure 2 and 3: indicate the long-time average in years, in brackets
L305: in this point indicate average precipitation and temperature of the vegetative season in 2020 and 2021
L310-311: indicate date of herbicides application or BBCH stage of milk thistle to whom herbicides were applied. Do you mean quizalofop-p-ethyl?
L312: do you mean 24 samples of milk thistle seeds?
L317-318: indicate date of harvesting or BBCH stage of milk thistle
L332: do not start a sentence with a number
L382: one-way ANOVA instead single factor
Figure 3 has no reference in the text
Author Response
Dear Sir or Madam,
thank you for your inspiring comments on this article. We also analysed other mycotoxins such as deoxynivalenol, zearalenone, ochratoxin A and the sum of T-2 and HT-2 toxins in milk thistle samples, where the sum of these two toxins was alarmingly high and therefore, we decided in this article to focus on the content of these mycotoxins using immunoaffinity columns and UPLC/MS-MS. Hereby we would like to draw attention to the insufficient legislative treatment of mycotoxins in medicinal plants.
Corrections are given in the form of revisions directly in the manuscript.
L166 I respect the remark that it is not appropriate to formulate the "impact of the year", but the "impact of climatic conditions in the year", but in terms of the character of the article, it is not constructive to include temperature and precipitation in the ANOVA.
L169-172 Given the length of the experiment, it is not constructive to correlate the course of temperatures and precipitation with the content of toxins.
Kind regards,

Reviewer 2 Report
The article: “Determination of T-2 and HT-2 Toxins in Seed of Milk Thistle Using Immunoaffinity Column by UPLC-MS/MS” validates a new methodology for the determination of T-2 and HT-2 mycotoxins by using immunoaffinity columns. In addition, studies the contamination of milk thistle seeds during two different periods (2020-2021) in the Czech Republic. The paper is well written and shows the problematic contamination of mycotoxins in this medicinal herb.
Some comments are indicated below:
-Lines 44 and 45: writhe the % next to the number: 2% and 1%.
-Line 125: How were the interlaboratory assays performed?
-Line 142: I think you wanted to write: Results of this study are shown…
-Lines 149-165: Change the paragraph to the appropriate format.
Discussion section:
The discussion is adequate and complete. It is argued that the climatology of both periods studied could have affected the mycotoxin content. I suggest that the authors find other studies conducted with mycotoxin contamination to support their hypothesis.
-Figures 2 and 3: The figures need to provide all the information necessary. I suggest modifying the legend. Also, where is the information provided by the author taken from? It is not mentioned in the text.
The paper contains a thorough bibliography revision. However, some of the references are outdated. I suggest the author (if possible) replace those references outdates with some actualized ones.
Author Response
Dear Sir or Madam,
Thank you for your inspiring comments on this article.
Figures 2 and 3 Long-term average precipitation and temperatures added directly to the legend. The data were obtained from a weather station located directly on the Agritec Plant Research Plc., where the experiment was established. The station is managed by the Czech Hydrometeorological Institute ID: O2SUMP01. GPS coordinates are added to the text.
I have tried to replace outdated references with newer sources, also mentioned in the form of revisions.
Kind regards,

Round 2
Reviewer 1 Report
Authors corrected the manuscript according to the comments. I have one more suggestion:
L293-294: Information related to meteorological station should be included in the Materials and Methods
Author Response
L293-294: Information related to meteorological station included in the Materials and Methods
